# A Sensitive, Cell-Based Assay for Measuring Low-Level Biological Activity of α-Amanitin

**DOI:** 10.3390/ijms242216402

**Published:** 2023-11-16

**Authors:** Reuven Rasooly, Paula Do, Xiaohua He, Bradley Hernlem

**Affiliations:** Foodborne Toxin Detection & Prevention Research Unit, Western Regional Research Center, Agricultural Research Service, United States Department of Agriculture, Albany, CA 94710, USA; paula.do@usda.gov (P.D.); xiaohua.he@usda.gov (X.H.); bradley.hernlem@usda.gov (B.H.)

**Keywords:** α-amanitin, qPCR, MTT, adenovirus, GFP, cell-based assays, food safety

## Abstract

α-Amanitin is one of the primary toxins produced by the poisonous mushroom genus, *Amanita*. Because it is odorless and tasteless, it is an important cause of death from the consumption of misidentified mushrooms. To study the thermal stability of α-amanitin, novel cell-based assays were developed to measure the toxin’s activity, based on the inhibition of RNA polymerase II by α-amanitin. First, an MTT–formazan cell viability assay was used to measure the biological activity of α-amanitin through the inhibition of cellular activity. This method can detect 10 μg/mL of α-amanitin in a time-dependent manner. Second, a more sensitive quantitative PCR approach was developed to examine its inhibition of viral replication. The new RT-qPCR assay enabled the detection of 100 ng/mL. At this level, α-amanitin still significantly reduced adenovirus transcription. Third, a simpler GFP expression-based assay was developed with an equal sensitivity to the RT-qPCR assay. With this assay, aqueous α-amanitin heated at 90 °C for 16 h or treated in the microwave for 3 min retained its biological activity when tested in HEK293 cells, but a slight reduction was observed when tested in Vero cells. Beyond detecting the activity of α-amanitin, the new method has a potential application for detecting the activity of other toxins that are RNA polymerase inhibitors.

## 1. Introduction

*Amanita phalloides* is one of the most poisonous mushrooms, causing most of the fatal human cases of mushroom poisoning worldwide. This species contains three main groups of toxins: amatoxins, phallotoxins, and virotoxins [1]. Amatoxins, especially α-amanitin, are mainly responsible for the toxic effects in humans. Poisonous mushrooms that contain amatoxins have no distinct odor or taste produced by the toxins and are often indistinguishable by color or appearance from edible mushrooms [2]. This is the main reason they are responsible for 95% of deaths from mushroom consumption worldwide [2,3,4]. Among the nine amatoxins found in several genera of poisonous mushrooms, the bicyclic octapeptide, α-amanitin, is the most studied and most abundant, with reported concentrations up to and exceeding 4 mg/g of α-amanitin in dry tissues [5,6]. 

Mechanistically, α-amanitin inhibits protein synthesis by selectively inhibiting RNA polymerase II-mediated transcription by directly interfering with the mobile elements in the catalytic center of RNA polymerase II (RNAP II), the bridge helix, and the trigger loop. α-Amanitin prevents RNAP II translocation after nucleotide addition [7,8], preventing the entry of the next NTP substrate [9]. The canine LD50 is 0.1 mg/kg of α-amanitin by body weight if administered intravenously, and death occurs 2 to 3 days after the dogs develop acute liver dystrophy. A late symptom of α-amanitin intoxication is kidney failure, from which a few dogs have died, though not before 7 days after intoxication. Humans are at least as sensitive as dogs and absorb the mycotoxin quickly since the consumption of a single mushroom seems to be sufficient to kill an adult [10]. After the consumption of *A. phalloides*, four main phases occur: (1) the lag stage or latency period, during which there are no symptoms, (2) the gastrointestinal (GI) phase, (3) subsidence of the GI phase while the liver and kidneys are attacked (hepatorenal phase), and (4) the multiorgan failure stage [6,11]. The second stage is characterized by bloody diarrhea, vomiting, nausea, and hematuria, generally occurring 6–24 h after ingestion [12]. During the third stage, GI symptoms improve, but the hepatic and renal functions are still deteriorating. In the last stage, patients lose liver and kidney function, which may lead to jaundice, hypoglycemia, and delirium. It has been estimated that the mortality rate is between 10–30% [13], and 20–79% of patients will develop chronic liver disease [14]. Survival is dependent on the amount of hepatic destruction, the ability of the remaining liver cells to regenerate, and the management of any complications [15]. General treatments normally include substantial hydration, prevention of toxin absorption, and the stabilization of vitals [16]. The use of liver transplantation has been effective in increasing the survival rate, but this treatment is costly, risky, and not readily available. [17]. Previously, five potential antidotal therapies (N-acetylcysteine, benzylpenicillin, cimetidine, thioctic acid, and silybin) for limiting the degree of hepatonecrosis have been tested, but none were effective in preventing hepatic injury after α-amanitin poisoning [18]. However, Garcia et al. showed that polymyxin B could potentially be used as a treatment. Polymyxin B binds to RNAP II in the same pathway as α-amanitin, preventing the toxin from binding to RNAP II [19]. Most recently, clustered regularly interspaced short palindromic repeats (CRISPR) screening was conducted in search of a treatment for α-amanitin. It was determined that STT3B and the N-glycan biosynthesis pathway are required for α-amanitin toxicity and that indocyanine green could potentially inhibit these pathways and be used as a treatment [20]. But for indocyanine green to be effective, it must be given as early as possible. There is an urgent need to develop detection methods for α-amanitin. Whatever treatment is being used will be most effective when given in the early stages of α-amanitin poisoning. 

Various analytical methods have been developed to identify and quantify poisonous α-amanitin toxins in biological fluids. These include capillary electrophoresis [21,22], liquid chromatography–mass spectrometry (LC–MS) [9,10,23,24,25,26], radioimmunoassays [27,28], lateral flow immunoassays (LFIA) [29], and enzyme-linked immunosorbent assays (ELISA) [30]. These commonly used detection methods can only detect the presence of amatoxins and not their activity. It has recently been reported by Sharma et al. [31], using high-performance liquid chromatography (HPLC), that α-amanitin is thermally degraded in mushrooms that have been subjected to common cooking methods, i.e., boiling, baking, grilling, and frying. Kaya et al. [32] reported that the remaining level of α-amanitin was only 5% after 6 h in boiling water. They cautioned, however, that although the toxin had been chemically degraded, the products might still have biological activity. Others [33] have disputed these findings, repeating the long-held view that α-amanitin is highly resistant to heat and other processes used in the consumption of mushrooms, i.e., cooking, freezing, and drying. To address this controversial issue, we chose kidney cell lines because of its high rate of protein synthesis and harnessed the ability of α-amanitin to inhibit RNA polymerase II. Two new assays, a quantitative RT PCR assay and a fluorescent protein inhibition assay, were developed to detect biologically active α-amanitin. The latter, simpler assay was used to examine the heat stability of α-amanitin.

## 2. Results

### 2.1. Detection of Biological Activity of α-Amanitin with an MTT–Formazan Cell Viability Assay

Since the immunoassays and chromatography methods do not distinguish between active and inactive α-amanitin, an MTT–formazan cell viability assay was tested as an initial candidate for detecting the biological activity of α-amanitin. Increasing concentrations of α-amanitin from 0.01 pg/mL to 10 μg/mL were incubated with Vero cells for 24, 48, and 72 h. In this assay, the samples were measured spectrophotometrically at 540 nm after the reduction of the yellow tetrazolium salt 3-(4,5-dimethylthiazol-2-yl)-2,5-diphenyltetrazolium bromide (MTT) to form the purple MTT–formazan that accumulates proportionally to the number of viable cells with cellular mitochondrial dehydrogenase activity. As shown in Figure 1, the addition of α-amanitin to Vero cells at toxin concentrations of 1 ng/mL and 100 ng/mL resulted in a significant increase in cellular metabolic activity and the accumulation of the water-insoluble purple formazan crystal after 24 h and 72 h. There was no linear dose–response relationship for the concentration of α-amanitin. As the concentration increased, there was an increase in the cellular enzyme activity, which overcame the effects of toxicity, but at a concentration of 10 μg/mL of α-amanitin, there was an inhibition of cellular activity in a time-dependent manner and damage to the cells beyond their limits of recovery. By comparison, 10 μg/mL of Shiga toxin group 2 (Stx2), a bacterial toxin that also inhibits protein synthesis, had a similar effect to α-amanitin at a concentration of 10 μg/mL. The low sensitivity of the MTT–formazan activity assay suggests that there is a need to develop more sensitive methods for analyzing the activity of α-amanitin. 

### 2.2. Validation of the qPCR Primers for Quantification of Adenovirus Genome Copy Number

The inhibitory effect of α-amanitin on dehydrogenase activity, as measured by the accumulation of MTT–formazan formed after the reduction of MTT, was only observed at 10 μg/mL. An alternate means of detection is the quantification of the inhibitory effect of α-amanitin on RNA polymerase II in HEK293 cells, the multi-subunit enzyme that transcribes DNA to mRNA in transduced cells. The inhibition of viral replication is a more direct measure of the activity of α-amanitin. The quantitative polymerase chain reaction (qPCR) primers were validated for the quantification of the adenovirus genome copy number by performing the qPCR assay on human adenovirus serotype 5 after they burst out from HEK293 cells during lysis, which kills the host cell, and after determining their titer with a plaque assay. This virus was serially diluted by a factor of five over a range of copy numbers from 5 × 10^7^/mL to 3 × 10^2^/mL. Figure 2a shows that there is linear correlation of R^2^ = 0.94 over a more than 4-log range between the increase in the adenovirus dilution ratio and the decrease in the viral CT value. Therefore, the primers designed to amplify 162 base pairs of the human adenovirus 5 gene can be used for the quantification of the adenovirus copy number. The qPCR product was sequenced and analyzed with agarose gel electrophoresis, and it was confirmed that the amplicons were 162 base pairs in length, as shown in Figure 2b. 

### 2.3. RT-qPCR for Evaluation of the Inhibitory Effect of α-Amanitin on mRNA Synthesis

To evaluate the inhibitory effect of α-amanitin, the reduction of adenovirus mRNA synthesis in HEK293 cells was quantitatively measured using reverse transcription quantitative real-time polymerase chain reaction (RT-qPCR) by adding decreasing concentrations of α-amanitin from 10 μg/mL to 100 pg/mL to HEK293 cells. After a 24 h incubation, the cells were transduced with the adenovirus at an MOI of 100. After an additional 24 h of incubation, RNA was isolated, and cDNA was synthesized. qPCR was used to quantitatively measure the fold change in adenovirus replication. To confirm that the qPCR products were indeed 162 base pairs, the qPCR product was analyzed using gel electrophoresis. Then, the amplicons were sequenced to confirm that they were not originating from the HEK293 cells but were from the human adenovirus type 5 DNA sequence range from base pair 101 to 262 of the accession number AC_000008.1 that was assigned to this adenovirus sequence. The target 162 base pair sequence is ***GTAGTGTG GCGGAAGTGT GA*** TGTTGCAA GTGTGGCGGA ACACATGTAA GCGACGGATG TGGCAAAAGT GACGTTTTTG GTGTGCGCCG GTGTACACAG GAAGTGACAA TTTTCGCGCG GTTTTAGGCG GATGTTGTAG TAAA ***TTTGGG CGTAACCGAG TAAG***. (The PCR primers are shown in bold/italics). The quantitative RT-qPCR analysis in Figure 3 shows that this RT-qPCR assay has a limited detection sensitivity of 100 ng/mL. At that concentration, α-amanitin significantly reduced adenovirus transcription via the inhibition of RNA polymerase II. The lower concentration of 1 ng/mL of α-amanitin appeared to increase the total adenovirus number, as demonstrated by the higher levels of viral DNA amplicons. However, because the standard error was large, this constitutes no significant difference compared to the control.

### 2.4. Green Fluorescent Protein in Transduced Cells for Detection of Biologically Active α-Amanitin

Kaya et al. [32], using HPLC, reported that the remaining level of α-amanitin was only 5% after 6 h in boiling water. Our lab has previously shown that microwave treatment degrades Shiga toxin and reduces its ability to inhibit protein synthesis [33]. To test if α-amanitin is heat sensitive, the activity assay was first made easier to perform. An adenovirus was generated that expressed the green fluorescent protein (GFP) gene under the control of the cytomegalovirus (CMV) immediate-early promoter. Decreasing concentrations of α-amanitin from 10 μg/mL to 100 pg/mL were added to HEK293 and Vero cell lines that had been heated for 16 h at 90 °C or microwave treated for 3 min. Then, both cell lines were transduced with an adenovirus that expressed the GFP at an MOI of 100, and the GFP fluorescence emission was measured at 24 h for HEK293 cells and 72 h for Vero cells to detect the inhibition of protein synthesis. The quantitative fluorescence emission assay results, represented in Figure 4a for HEK293 cells and Figure 5a for Vero cells, show that the sensitivity of the assay in HEK293 cells, which contain the E1A gene that is essential for adenovirus replication, is 100 ng/mL, similar to the sensitivity of the RT-qPCR assay. The result in Figure 5a shows that the sensitivity of the assay in Vero cells, which are unable to support adenovirus replication, is 1 μg/mL, which is 100 times less than HEK293 cells. By using transduced HEK293 cells, this fluorescent protein inhibition assay is as sensitive as qPCR. To test if heat treatment reduces the biological activity of α-amanitin, decreasing concentrations of α-amanitin that had been heated for 16 h at 90 °C or microwave treated for 3 min were added to transduced HEK293 and Vero cells. The results in Figure 4 show that α-amanitin heated at 90 °C for 16 h (b) or microwave treated for 3 min (c) did not reduce the biological activity of α-amanitin when tested in HEK293 cells. There are no statistically significant differences between the unheated α-amanitin and those two heated treatments when tested in HEK293 cells. However, Figure 5b shows that the heat treatment at 90 °C for 16 h slightly changes the biological activity of α-amanitin when tested in Vero cells.

## 3. Discussion

In this study, it was demonstrated that the biological activity of α-amanitin remains after a heat treatment. This question was raised after a recent publication by Sharma et al. [31] reported that α-amanitin is thermally degraded in mushrooms subjected to common cooking methods, i.e., boiling, baking, grilling, and frying. They found the greatest reduction in boiled mushrooms but noted that this was likely to be partly due to the toxin’s high water solubility. Earlier work by Kaya et al. [32] examined the stability of α-amanitin in water and methanol stored in sealed vials under freezing, refrigeration, room temperature, and boiling. In the latter case, they reported that the level of α-amanitin remaining after 6 h in boiling water was only 5%. However, they cautioned that although the toxin had been chemically degraded (based on HPLC separation), the products might still have biological activity. Previous methods that have been established to detect α-amanitin had a limited detection of 0.5–1.5 ng/mL for LC–MS [25], 10 ng/mL for LFIA [29], and around 1 ng/mL for ELISA [30]. However, such commonly used detection methods can only detect the presence of amatoxins but do not distinguish between active and inactive α-amanitin. 

In this present work, the goal was the development of cell-based activity assays to reexamine the question of the effect of thermal treatment on the activity of α-amanitin. First, this was determined by using the colorimetric MTT–formazan cell viability assay. The results in Figure 1 showed that α-amanitin at a concentration of 10 μg/mL had a similar time-dependent effect as Shiga toxin group 2 (Stx2), a bacterial toxin that also inhibits protein synthesis. The sensitivity of the MTT assay was not impressive, so an alternative means of detection was developed based on the inhibitory effect of α-amanitin on RNA polymerase II in HEK293 cells and utilizing a qPCR assay, a 100-fold improvement in the detection sensitivity was observed. These two assays measure different outcomes of α-amanitin activity: loss of cell viability and inhibition of adenovirus replication. Unlike the Vero cell line, HEK293 cells contain the E1A region that is essential for adenovirus replication. Therefore, this cell line was used for RT-qPCR to quantitatively measure the fold change in adenovirus replication in transduced HEK293 cells treated with decreasing concentrations of α-amanitin. As demonstrated by the levels of viral DNA amplicons in Figure 3, at a high concentration of α-amanitin, the toxin’s cytotoxicity effect damaged the cells and reduced virus transcription. However, the lower concentration of 1 ng/mL α-amanitin appeared to increase the total adenovirus number, as demonstrated by the higher levels of viral DNA amplicons. However, because the standard error was large this constitutes no significant difference compared with control. Similar results were shown in Figure 1 with cytotoxic effects at a concentration of 10 μg/mL α-amanitin, damaging the cells beyond their limit of recovery, while a lower concentration of 1 ng/mL and 100 ng/mL had an opposite effect causing an increase in cellular enzyme activity that overcame the cytotoxicity effect. It is important to be aware of this paradoxical effect of α-amanitin. For example, to deliver an effective cytotoxic dose of α-amanitin to prevent protein synthesis, where α-amanitin is used in cancer treatment in combination with monoclonal antibodies specific to surface antigens present on target tumor cells, this effect must be considered. With the observation by qPCR that α-amanitin inhibits adenovirus replication in cells, the assay was simplified by using adenoviral expression of the green fluorescent protein (GFP) gene. The quantitative fluorescence emission results presented in Figure 5 showed that HEK293 cells, which contain the E1A gene that is essential for adenovirus replication, increased the sensitivity of the assay by 10-times compared to Vero cells which are unable to support adenovirus replication. 

This assay was used in HEK293 and in Vero cells to address the controversial issue of α-amanitin heat sensitivity. The results in both cell lines showed that heat treatment did not reduce the biological activity of α-amanitin. These results are consistent with the view that α-amanitin is thermally stable. In contrast, the results reported by Sharma et al. [31] and Kaya et al. [32] did not measure toxin activity. In both cases, they used HPLC to quantify α-amanitin after heat treatment in mushroom materials and purified toxin in sealed vials, respectively. In both cases, they found a loss or degradation of the toxin molecule. At least some of the loss in the boiled mushrooms could be explained by the solubility of the toxin in the cooking water. Kaya et al. [32] cautioned that although they observed only 5% of the toxin remaining after 6 h in boiling water, the activity may remain. That appears to be the case in that here, it was found that there was little to no loss of biological activity after a heat treatment at 90 °C for 16 h. These results also found the activity of α-amanitin to be stable after microwave treatment, a process commonly used to heat foods. However, it has been shown to degrade the thermally stable Shiga toxin by breaking up its A subunit using an apparently nonthermal mechanism [33]. The detection of Shiga toxin group 2 (Stx2) may suggest that beyond measuring low levels of activity in α-amanitin, the technology has the potential for detecting low levels of activity of other RNA polymerase II inhibitor toxins, including *Kluyveromyces lactis-*produced zymocin [34], Thiolutin, which is produced by *Streptomycetes luteosporeus* [35], and Tagetitoxin phytotoxin [36], and for other food safety applications.

## 4. Materials and Methods

### 4.1. Cell Culture

HEK293 and Vero cells that were purchased from the American Type Culture Collection (ATCC, Manassas, VA, USA) were maintained in Dulbecco’s Modified Eagle Medium (DMEM, thermo fisher scientific, Waltham, MA, USA) containing 4.5 g/L of D-glucose, 110 mg/L of sodium pyruvate, 200 mM of L-glutamine, 10% fetal bovine serum, 100 units/mL each of penicillin and streptomycin, and 0.25 µg/mL of Amphotericin B. Cells were harvested by trypsin when they reached about 90% confluency (Thermo fisher scientific, Waltham, MA, USA).

### 4.2. MTT Assay

The MTT (3-[4,5-dimethylthiazol-2-yl]-2,5 diphenyl tetrazolium bromide) assay is based on the conversion of MTT into formazan crystals by oxidoreductase and dehydrogenase enzymes in cells, which is generally associated with mitochondrial activity. Briefly, Vero cells were plated on 96-well plates at a density of 1 × 10^5^ cells in 100 µL of DMEM medium per well. Cells were incubated overnight to allow cell attachment to the plate. α-Amanitin was added to each well and incubated for 72 h at 37 °C in a 5% CO_2_ incubator. Twenty-five µL of MTT in PBS at a concentration of 2 mg/mL was added to each well. Plates were incubated at 37 °C for 4 h, and the medium was removed. A total of 100 μL of dimethyl sulfoxide (DMSO) was added to each well, and plates were read at 540 nm.

### 4.3. Quantitative PCR

HEK293 cells were plated on 96-well plates with 1 × 10^4^ cells in 100 μL of medium per well. Cells were incubated overnight at 37 °C in a 5% CO_2_ incubator to allow the cells to attach to the plate. After 24 h, decreasing concentrations from 10 μg/mL to 1 ng/mL of α-amanitin were added to each well. After 24 h, the treated cells were transduced with the adenovirus at a multiplicity of infection (MOI) of 100. After an additional 48 h, the total RNA was prepared using the RNeasy kit as described by the manufacturer (Qiagen, Hilden, Germany). The isolated RNA was used to synthesize first-strand cDNA using reverse transcription. The reaction was performed with 1.0 mg of total RNA, oligo-dT primers, and reverse transcriptase Superscript II (Life Technologies, Carlsbad, CA, USA) in a total volume of 25 μL. RNaseH was used to degrade the RNA after the reaction. An Eppendorf Mastercycler ep Realplex thermal cycler (Eppendorf, Hauppauge, New York, NY, USA) was used for the quantitative PCR analysis using Brilliant SYBR Green qPCR master mix reagents (Agilent Technologies, Santa Clara, CA, USA). The reaction mixture comprised 1 μL of cDNA and 0.25 µmol of primers in a total reaction volume of 20 µL. The reaction sequence began with a 3 min denaturation at 95 °C, after which 40 thermal cycles were performed comprising a 30 s denaturation at 94 °C, a 30 s annealing at 56 °C, and a 1 min extension at 72 °C. Upstream and downstream primers for genes of interest were as follows: The human housekeeping gene, GAPDH, which controls for variance among samples and ensures that the number of cells per sample and the efficiency of cDNA synthesis are controlled for in the final data, was purchased from Bio-Rad (Hercules, CA, USA). The primers used for the adenovirus genome were the forward primer 5′-GTAGTGTGGCGGAAGTGTGA-3′, corresponding to nucleotides 103 through 123 of human adenovirus 5, Sequence ID: AC_000008.1, and the reverse primer 5′-CTTACTCGGTTACGCCCAAA-3′, corresponding to nucleotides 242 through 262. Confirmation of the correct amplification of the sequence of interest using the primer pair was achieved with agarose gel electrophoresis (1% SeaKem 3:1). Gels were stained with GelRed (Phenix Research, Candler, NC, USA) and visualized with UV transillumination. A single product band of the expected size was observed, and the fold change was calculated with the 2^−ΔΔCT^ method. We cloned the adenovirus 162 bp PCR-amplified amplicon and sent the material to the company Genewiz Inc. for sequencing. They use the Sanger DNA sequencing method as described previously [37].

### 4.4. Plaque Assays for Titration of the Adenovirus

Six 35 mm tissue culture plates were seeded with HEK293 cells. The cells were incubated at 37 °C in a CO_2_ incubator until the cells were 90% confluent. Serial dilutions of the adenovirus stock were made in Dulbecco’s Modified Eagle Medium (DMEM) supplemented with 2% FBS. The diluted virus was added to the cells. After 2 h, the medium was removed and replaced with 1X Modified Eagle Medium and 1% SeaPlaque agarose (FMC). The agar overlay was added to keep the virus localized after the cells had lysed. After 5 days, plaques were visible and were counted for titer determination after 7 days.

### 4.5. Heat Treatment of α-Amanitin

Two experiments were performed to determine the effect of heat on the activity of α-amanitin. An aqueous solution of 1 mg/mL of α-amanitin was prepared. Aliquots of 40 μL of this solution were transferred to microcentrifuge tubes and treated with heat either in an Eppendorf dry heating block (Government Scientific Source, Inc., Reston, VA, USA) or a microwave oven. In the first case, the thermal block was held at 90 °C for 16 h. In the second case, the tube was placed next to a beaker of water initially containing 60 mL of water. After three minutes of microwave treatment, the rapid rotation of polar water molecules created internal heat that evaporated 40 mL from the beaker. After cooling, the samples were analyzed for the activity of α-amanitin.

### 4.6. Generation of Adenoviral-Expressed GFP Gene

As previously described [38,39], briefly, HEK293 cells were co-transfected with pJM17, a plasmid constructed with an entire copy of the Ad5 adenovirus genome, a 4.4-kb sequence encoding for ampicillin and tetracycline resistance, and an adenoviral shuttle plasmid constructed from sequences encoding the CMV promoter containing the 750-bp GFP gene flanked by the adenovirus E1 sequences. After 10 days of incubation, cytopathogenesis caused the transfected cells to become nonadherent, detach from the plate, and appear round. Fluorescence microscopy was used to analyze the cells for the expression of GFP. A single plaque displaying the expression of the adenovirus vector and GFP (Ad-GFP) was selected and amplified. The level of GFP expression in transduced cells was quantified using the Synergy HT Multi-Detection Microplate Reader (BioTek, Winooki, VT, USA). The excitation was at a wavelength of 485 nm using a 485/20 bandpass filter, and the fluorescence was measured at a wavelength of 528 nm using a 528/20 bandpass filter.

### 4.7. Fluorescent Ad-GFP Assay

HEK293 and Vero cells were plated on black 96-well plates (Greiner 655090 obtained from Sigma, Burlington, MA, USA) with 3 × 10^4^ cells in 100 μL of medium per well. Cells were incubated overnight at 37 °C in a 5% CO_2_ incubator to allow the cells to attach to the plate. After 24 h, decreasing concentrations from 10 μg/mL to 100 pg/mL of α-amanitin heat treated at 90 °C for 16 h or microwave treated for 3 min were added to each well. Then the α-amanitin-treated cells were transduced with the GFP-expressing adenovirus at an MOI of 100. The GFP fluorescence emission was measured at 24 h for the HEK293 cells and 72 h for the Vero cells to detect the inhibition of protein synthesis. To improve the signal-to-noise ratio, the autofluorescent DMEM media was removed. The fluorescence emission at 528/20 nm was measured with excitation at 485/20 nm using the Synergy HT Multi-Detection Microplate Reader (BioTek, Winooki, VT, USA).

### 4.8. Statistical Analysis

Statistical analysis was performed using SigmaStat 3.5 for Windows (Systat Software, San Jose, CA, USA). Multiple comparisons of the treated α-amanitin or control were made. One-way analysis of variance (ANOVA) was used to compare the increasing concentrations of α-amanitin, heat-treated α-amanitin, and the untreated control media. The experiments were repeated at least three times, and results with *p* < 0.05 were considered statistically significant.

## 5. Conclusions

The ability to measure and quantify the activity of α-amanitin is vital to determine the actual potential to cause harm to health. HPLC, which only detects the presence of amatoxins and not their activity, shows a substantial degradation and loss of detectable α-amanitin after heating in boiling water. Activity methods were developed in this study to detect and quantify the inhibition of RNA polymerase II by α-amanitin using an MTT–formazan cell viability assay, a qPCR measurement of adenovirus genome copy number, and a fluorescence measurement of GFP-adenoviral expression. The latter two methods were similarly sensitive, but the assay measuring levels of GFP expression in transduced HEK293 cells was the simplest and easiest to perform. This novel method for measuring low levels of 100 ng/mL toxin activity has other potential applications, such as for determining the activity level of α-amanitin used in targeted cancer treatments.

## Figures and Tables

**Figure 1 ijms-24-16402-f001:**
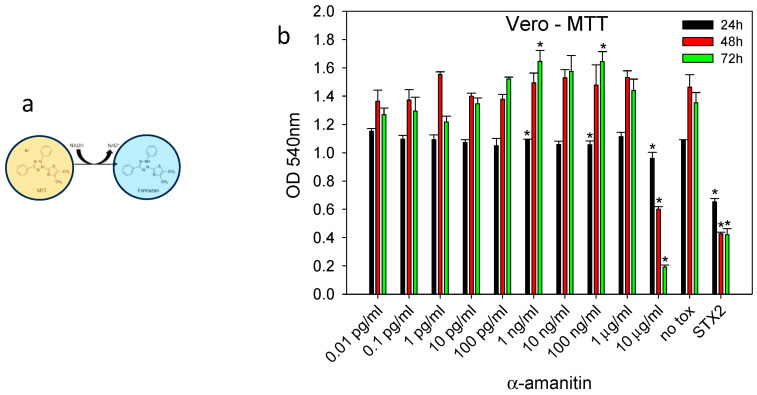
(**a**) MTT is reduced in metabolically active Vero cells to form an insoluble purple formazan product. (**b**) Decreasing concentrations of α-amanitin were incubated with Vero cells for 24, 48, and 72 h. A control with no toxin and a comparison with 10 µg/mL of Stx2 were included. The accumulation of MTT–formazan produced after the reduction of the yellow MTT to purple formazan was monitored spectrophotometrically and reported as a percentage of cell survival. (*) indicates significant differences (*p* < 0.05) between spiked and untreated control media. Error bars represent standard errors.

**Figure 2 ijms-24-16402-f002:**
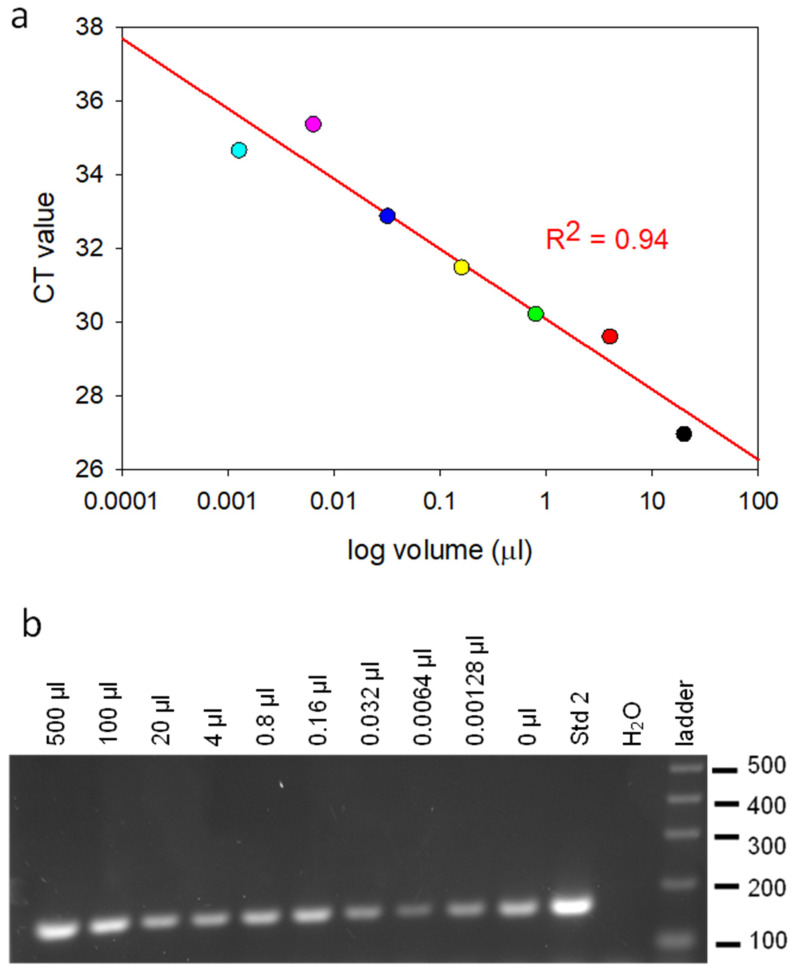
The quantitative polymerase chain reaction (qPCR) primers were validated for quantification of adenovirus genome copy number by performing qPCR assay on pure adenoviruses that were diluted by a factor of five over a range of copy numbers from 5 × 10^7^/mL to 3 × 10^2^/mL (**a**). The 162 base pair qPCR product was confirmed with gel electrophoresis (**b**).

**Figure 3 ijms-24-16402-f003:**
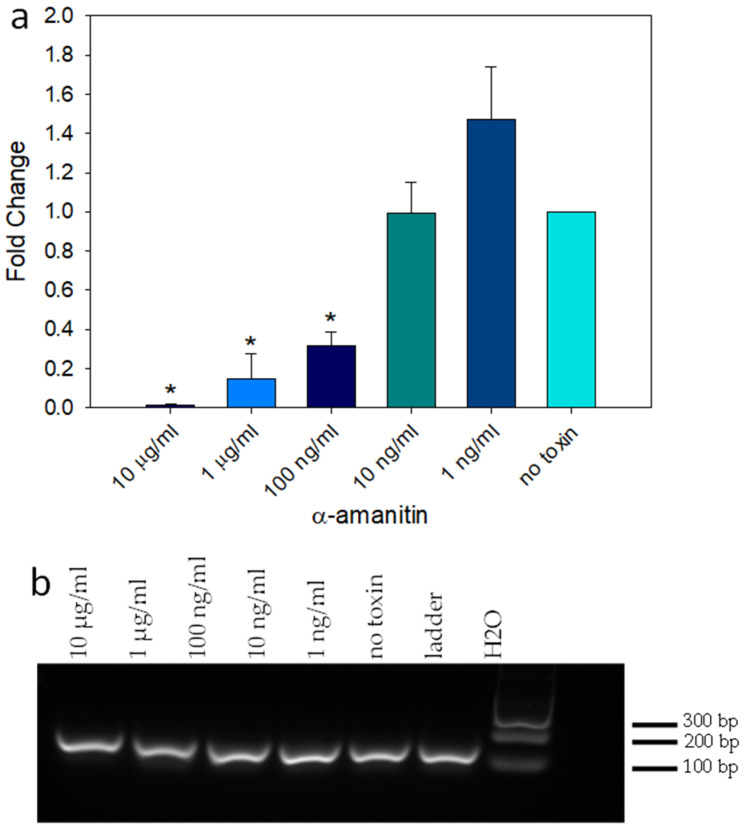
Reduction of mRNA synthesis by α-amanitin. Decreasing concentrations of α-amanitin and a control without any toxin were incubated with HEK293 cells for 24 h, then transduced with adenovirus at an MOI of 100. After incubation for 24 h, RT-qPCR was performed, and the fold change was calculated. (*) indicates significant differences (*p* < 0.05) between spiked and untreated control media. Error bars represent standard errors (**a**). RT-qPCR cDNA amplification of 162 bp product from adenovirus (**b**).

**Figure 4 ijms-24-16402-f004:**
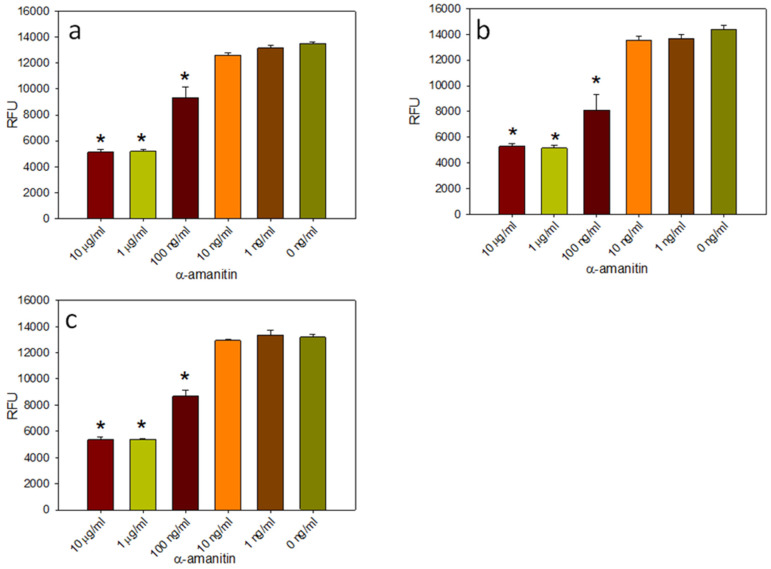
Decreasing concentrations of untreated α-amanitin (**a**), heated at 90 °C for 16 h (**b**), or microwave treated (**c**) were incubated with HEK293 cells for 24 h, then transduced with an adenovirus at an MOI of 100. After incubation for 24 h, GFP fluorescence emission was measured to detect the inhibition of protein synthesis. (*) indicates significant differences (*p* < 0.05) between α-amanitin treated cells and untreated control HEK293 cells. Error bars represent standard errors.

**Figure 5 ijms-24-16402-f005:**
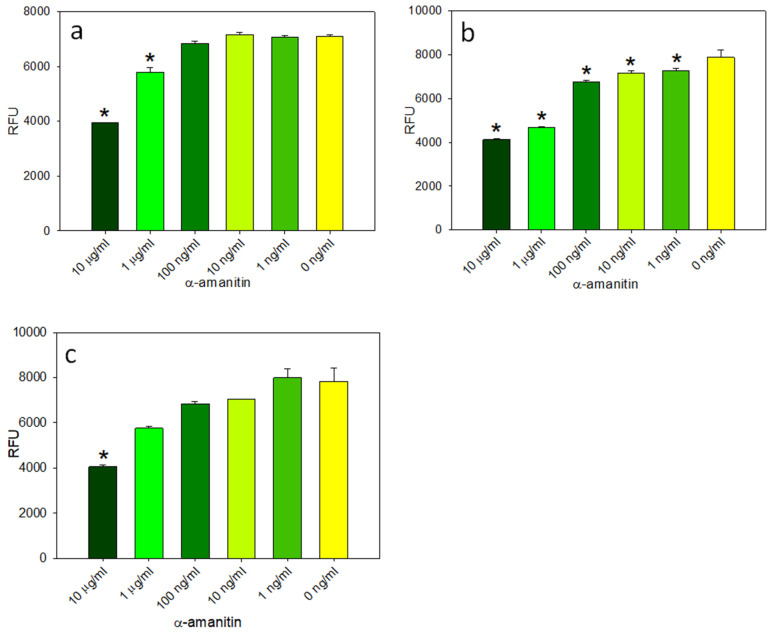
Decreasing concentrations of untreated α-amanitin (**a**), heated at 90 °C for 16 h (**b**) or microwave treated (**c**), were incubated with Vero cells for 24 h, then transduced with adenovirus at MOI of 100. After incubation for 24 h, GFP fluorescence emission was measured to detect the inhibition of protein synthesis. (*) indicates significant differences (*p* < 0.05) between α-amanitin treated cells and untreated control Vero cells. Error bars represent standard errors.

## Data Availability

The data that support the findings of this study are available from the corresponding author upon reasonable request.

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
