# Peer review of "A Sensitive, Cell-Based Assay for Measuring Low-Level Biological Activity of α-Amanitin"

_ijms, 2023, doi:10.3390/ijms242216402_

Round 1

Reviewer 1 Report

Comments and Suggestions for Authors

The article “A sensitive cell-based assay for measuring low-level biological activity of α-amanitin” focuses on studying α-amanitin’s thermal stability, and developing a novel method for measurement of low levels of toxin activity, as well as the activity of toxin after heat treatment.

This paper may be suitable for publication following major revision. Authors should answer and resolve the following queries.

 1. In lines 46-47 is written, „α-amanitin is the most studied and abundant, with concentrations of 2-3 mg of α-amanitin per kilogram of dry tissue [5].“ Please, check this date. Also, the reference is from 1979, are there any newer studies?

 2. Some references as references 5,11 are too old.

 3. Why are Vero and HEK cells used as a model system? Why not some hepatic cell line? Please, explain.

4.  Line 156 CO2 incubator – please, put 2 in subscript

5. Why is MTT done only on Vero cells, not HEK cells? Why is not checked activity of heat-treated α-amanitin?

 6. In lines 213- 215 is written “As shown in Figure 1, the addition of α-amanitin to Vero cells at toxin concentrations ranging from 10 pg/ml to 1 μg/mL resulted in a significant increase in cellular metabolic activity and accumulation of the water-insoluble purple formazan crystal after 72 h.”

 However, there are no significant differences between spiked and unspiked control media. Therefore, this cannot be concluded.

8. In lines 278-279 is written ”The quantitative RT qPCR analysis in Figure 3 shows that this RT qPCR assay is 100 times more sensitive than the MTT-formazan cell viability assay.”

How could this be concluded if the MTT assay was done at different cell lines? Also, different incubation time was applied?

 9. In lines 281-283 is written “However, the lower concentration of 1ng/mL α-amanitin significantly increased the total adenovirus number, as demonstrated by the higher levels of viral DNA amplicons.”

But there is no significant difference compared with control

10. Figure 2 is missing the standard ladder label

11. In Figure 2 A and Figure 3 B, letters are not fully visible 

12. Figure 3 – It is not clear, labels too much on the left side

14. Under Figures 4b, 4c, 5b, and 5c should be written which exact sample is used, not only “α -amanitin”

15. In line 324 is written, “However, Figure 5(b) shows that heat treatment at 90°C for 16h slightly reduced α- amanitin biological activity when tested in Vero cells.”

How is this compared? Are there statistical differences?

16. Somewhere is written HEK somewhere HEK293, it should be always written in the same way.

17. In line  356 “LCMS” should be replaced with LC-MS.

 18. In the first paragraph of the discussion is two times repeated „Kaya et al. [31] reported that the remaining level of α-amanitin was only 5% after 6 hours in boiling water.  However, they indicated that although the toxin had been chemically degraded (based on  HPLC separation) the products might still have biological activity“

Comments on the Quality of English Language

Minor editing of English language required

Reviewer 2 Report

Comments and Suggestions for Authors

The topic is interesting and within the scope of this journal. 

In this paper, the authors aim to develop a new method to ensure α-amanitin thermal stability in vitro for measuring its activity at low levels by using an RNA polymerase II inhibitor. Overall, the manuscript is suitable for publication after minor modifications:

-Page 1

line 16: when mentioning low concentrations it should be specified which concentration.

-Page 3

line 125: Regarding total RNA used for PCR assay, it should be specified from the authors (could be in a table) the starting amount detected in the samples (total ng/mL) and especially purity ratios 260/280, 260/230 of each sample.

-Did the authors design the primers for PCR analysis? And with which database? (If there is any)

-Please check the caption of each figure. Caption should be correctly placed under the figures and they don’t appear well in the text.

Reviewer 3 Report

Comments and Suggestions for Authors

The article is interesting and deals with an important issue. I have a few comments about the article.

1. Abstract – according to the instructions, the size of the Abstract should be a maximum of 200 words. The number of words in the Abstract in the submitted article significantly exceeds the prescribed range, so it needs to be significantly shortened. Please limit the Abstract only to facts corresponding to the results obtained from your own research.

2. In the second paragraph of the Introduction, from line 49 to line 83, the text is very confusing. It is necessary to divide this long paragraph into several appropriate paragraphs according to thematic focus and relevant citations.

3. At the end of the Introduction section, I recommend clearly formulating the Aim of the research and also stating the examined hypothesis (or hypotheses) that was verified. The sub-goals are partially mentioned in the Discussion on lines 345-356 and also on lines 363-364.

4. All pictures 1, 2, 3 4 are out of order. Above all, the descriptions under the pictures are confusing. Some explanatory comments are better placed in the body of the article. Organize the relevant description under the image according to the Template and according to the instructions for authors.

5. Correct minor formal errors and inaccuracies, e.g. line 339 the space between the number and the unit "for 16h (B)".

6. Conclusion, focus only on presentations of the results of own research and experiments. In my opinion, it is inappropriate to cite authors and refer to their research and conclusions in Conclusions.

Round 2

Reviewer 1 Report

Comments and Suggestions for Authors

Authors significantly improved the manuscript titled " A sensitive cell-based assay for measuring low-level biological 2 activity of α-amanitin".

 However, it should be improved before being accepted in a highly renowned journal such as Int. J. Mol. Sci. 

Figure 2 is missing by using which cell line are these results obtained.

In the 251 line is the first time written  “multiplicity of infection (MOI)” although the abbreviation was previously used. Please, the first time when mentioning some term introduce an abbreviation. 

In lines 298-300 is written “The quantitative fluorescence emission assay results represented in Figure 4a for HEK293 cells and Figure 5a for Vero cells show that the sensitivity of the assay in HEK293 cells, which contain the E1A gene that is essential for adenovirus replication, is 100 ng/mL, like the sensitivity of RT qPCR assay.”  There is no need for repeating in lines 303-305 “By using transduced HEK293 cells, this fluorescent protein inhibition assay is as sensitive as qPCR and 100 times more sensitive than the MTT-formazan cell viability assay.” 

In my opinion, MTT should be performed on HEK293 cells, also. It is not recommended to compare results obtained by MTT assay on Vero cells and results obtained by RT qPCR after treatment of HEK293 cells. Response on different cell lines could vary significantly. 

For example, in lines 301-303 is written “ The result in Figure 5a shows that the sensitivity of the assay in Vero cells which are unable to support adenovirus replication is 1μg/mL, which is 10 times less than HEK293 cells.” 

It is not clear why it is believed that the results of the MTT assay obtained on Vero cells would be the same as on HEK293 cells

It is not clear if there are statistically significant differences between the activities of untreated α-amanitin, heated at 90°C for 16 h  or microwave-treated.

In lines 358-362 is written” As demonstrated by the levels of viral DNA amplicons Figure 2, at a high concentration of α-amanitin, the toxin’s cytotoxicity effect damaged the cells and reduced virus transcription. However, the lower concentration of 1ng/mL α-amanitin had the opposite effect and significantly increased the total virus number in HEK293 cells.”

Please, check this statement. 

Comments on the Quality of English Language

Minor editing of English language required.

Author Response

We answered the questions posed by Reviewer 1 and would like to thank Reviewer for the constructive comments. We feel that the manuscript has greatly benefited as a result.

Authors significantly improved the manuscript titled " A sensitive cell-based assay for measuring low-level biological 2 activity of α-amanitin". 

 However, it should be improved before being accepted in a highly renowned journal such as Int. J. Mol. Sci.  Figure 2 is missing by using which cell line are these results obtained. 

No cell lines were used to generate the data shown in Figure 2. The results were of adenovirus after they burst out from HEK293 cells during lysis which kills the host cell and after determining their titer by plaque assay. To clarify this point, we added this point to the text and also change the figure legend to state: “The quantitative polymerase chain reaction (qPCR) primers were validated for quantification of adenovirus genome copy number by performing qPCR assay on pure adenovirus that were diluted by a factor of five over a range of copy numbers from 5x107/mL to 3x102/mL (a). The 162 base pair qPCR product was confirmed by gel electrophoresis (b). 

In the 251 line is the first time written “multiplicity of infection (MOI)” although the abbreviation was previously used. Please, the first time when mentioning some term introduce an abbreviation. 

As suggested, we have introduced the abbreviation for MOI on line 116 and deleted it from line 251.  

In lines 298-300 is written “The quantitative fluorescence emission assay results represented in Figure 4a for HEK293 cells and Figure 5a for Vero cells show that the sensitivity of the assay in HEK293 cells, which contain the E1A gene that is essential for adenovirus replication, is 100 ng/mL, like the sensitivity of RT qPCR assay.” There is no need for repeating in lines 303-305 “By using transduced HEK293 cells, this fluorescent protein inhibition assay is as sensitive as qPCR and 100 times more sensitive than the MTT-formazan cell viability assay.” 

As suggested, we have deleted the repeated lines at 303-305.  

In my opinion, MTT should be performed on HEK293 cells, also. It is not recommended to compare results obtained by MTT assay on Vero cells and results obtained by RT qPCR after treatment of HEK293 cells. Response on different cell lines could vary significantly. For example, in lines 301-303 is written “ The result in Figure 5a shows that the sensitivity of the assay in Vero cells which are unable to support adenovirus replication is 1μg/mL, which is 10 times less than HEK293 cells.”  It is not clear why it is believed that the results of the MTT assay obtained on Vero cells would be the same as on HEK293 cells. 

As suggested, we have edited the manuscript to remove the comparison (at line 16 and line 308) between the MTT assay on Vero cells and results obtained by alternate means of detection; GFP expression-based assay and RT qPCR after treatment of HEK293 cells, reporting only the levels of sensitivity.

It is not clear if there are statistically significant differences between the activities of untreated α-amanitin, heated at 90°C for 16 h or microwave-treated. 

The asterisks in figure 5 indicate instances where we found statistically significant activities of α-amanitin (untreated and treated) when compared to the unspiked controls. Figure 5 shows that there are no statistically significant differences between unheated α-amanitin and microwave heat treated when tested in Vero cells. However, as indicated by asterisks there are statistically significant differences between the activities of α-amanitin heated at 90°C for 16 h and unheated α-amanitin or microwave treated α-amanitin when tested in Vero cells. Even more the results in figure 4 show that unheated α-amanitin (a) α-amanitin heated at 90°C for 16 h (b) or microwave heat treated for 3 minutes (c) did not reduce α-amanitin biological activity. There are no statistically significant differences between unheated α-amanitin and those two heated treatments when tested in HEK293 cells. For clarification we added this point to the text.

In lines 358-362 is written” As demonstrated by the levels of viral DNA amplicons Figure 2, at a high concentration of α-amanitin, the toxin’s cytotoxicity effect damaged the cells and reduced virus transcription. However, the lower concentration of 1ng/mL α-amanitin had the opposite effect and significantly increased the total virus number in HEK293 cells.” Please, check this statement. 

We corrected the statement in lines 358-362. As demonstrated by the levels of viral DNA amplicons Figure 3, at a high concentration of α-amanitin, the toxin’s cytotoxicity effect damaged the cells and reduced virus transcription. However, the lower concentration of 1ng/mL α-amanitin appeared to increase the total adenovirus number, as demonstrated by the higher levels of viral DNA amplicons. However, because the standard error was large this constitutes no significant difference compared with control.

 We thank the reviewers for their constructive comments.

Sincerely,

Reuven Rasooly, Ph. D.